# The Effect of Yearly-Dose Vitamin D Supplementation on Muscle Function in Mice

**DOI:** 10.3390/nu11051097

**Published:** 2019-05-17

**Authors:** Alan Hayes, Emma Rybalka, Danielle A. Debruin, Erik D. Hanson, David Scott, Kerrie Sanders

**Affiliations:** 1Institute of Sport and Health, Victoria University, Melbourne 3011, Australia; emma.rybalka@vu.edu.au (E.R.); danielle.debruin@live.vu.edu.au (D.A.D.); edhanson@email.unc.edu (E.D.H.); 2Australian Institute for Musculoskeletal Sciences (AIMSS), Melbourne 3021, Australia; david.scott@monash.edu (D.S.); ksanders@unimelb.edu.au (K.S.); 3Department of Medicine—Western Health, Melbourne Medical School, The University of Melbourne, Melbourne 3021, Australia; 4Department of Exercise and Sport Science, University of North Carolina at Chapel Hill, Chapel Hill, NC 27599, USA; 5School of Clinical Sciences at Monash Health, Monash University, Melbourne 3800, Australia

**Keywords:** cholecalciferol, vitamin D, muscle function, muscle fatigue

## Abstract

Supplementation with vitamin D helps to alleviate weakness and fatigue seen with deficiency. However, large bolus doses appear to worsen the risk of falls. Whether this occurs as a direct result of muscle weakness is currently unknown. Thus, the aims of this study were to examine the muscle function following administration of high doses of vitamin D. Given the safety issues associated with bolus doses, experiments were conducted on C57BL6 mice. Mice at eight weeks of age with otherwise normal levels of vitamin D were supplemented for four weeks with a high dose (HIGH; *n* = 12) of vitamin D (20000 IU/kg food) designed to provide a year’s worth of vitamin D. These mice were compared to another group who received that same yearly dose in a single bolus i.p. injection (YEAR; *n* = 12). Mice provided with standard mouse chow, which contained 1000 IU/kg food, and injected with the vitamin D vehicle were used as controls (CON; *n* = 16). Force and fatigue properties of hind limb fast- and slow-twitch muscles were measured. CON animals ingested vitamin D consistent with typical human supplementation. HIGH animals consumed significantly more food than the CON animals, such that they ingested more than a year’s worth of vitamin D in four weeks. Despite this, there were few differences in the muscle function compared with CON. YEAR animals demonstrated lower absolute and relative forces in both muscles compared to the HIGH animals, as well as lower force during fatigue and early recovery. Large bolus doses of vitamin D appear to have detrimental effects on the skeletal muscle function, likely being a contributor to increased risk of falls observed with similar doses in humans. Mice ingesting the same amount over four weeks did not demonstrate the same deleterious effects, suggesting this may be a safe way to provide high vitamin D if required.

## 1. Introduction

Vitamin D deficiency (defined as serum 25-hydroxyvitamin D levels below 50 nmol/L) affects approximately one-quarter of the Australian population [1,2]. Low vitamin D has been associated with muscle weakness and fatigue [3], with a subsequent higher rate of falls and fractures [4,5,6]. However, subsequent vitamin D supplementation can successfully reverse these effects [7,8,9], with vitamin D levels greater than 60 nmol/L being associated with a 23% falls reduction [10]. Indeed, older women have a reduced risk of falling with moderate doses (ranging from 1600 to 3200 IU/day) of vitamin D [11]. Given the well-established role of vitamin D (and calcium) in decreasing the risk of falls and fractures, with no apparent side effects [12], it is easy to see why vitamin D supplementation is recommended by health professionals.

However, once again the old adage of “too much of a good thing is bad for you” rings true for vitamin D supplementation. In a landmark study [13], participants in the VitalD study (2256 community dwelling women aged 70 years or over and considered to be at risk of sustaining falls or fractures) were provided with 500,000 IU cholecalciferol, or placebo, orally each autumn or winter for three to five years, with the intention of rapid increase in vitamin D levels which could be sustained throughout the year, to decrease the rate of falls and fractures. Unexpectedly, participants receiving the annual dose of cholecalciferol demonstrated an increased risk of 15% of experiencing falls and 26% increased risk of fractures. This was particularly prevalent in those women in which serum 25(OH) vitamin D increased beyond 100 nmol/L [13]. This supported a study in which intramuscular injections of a yearly dose of ergocalciferol, 300,000 IU, were considered to have an increased fracture risk, at least in older women [14].

Importantly, two more recent studies have supported the original study by Sanders et al [13]. Bischoff-Ferrari et al. [10] conducted a double blinded, single centre trial and reported a greater than five-fold increased falls risk when serum levels of 25(OH)D rose above 100 nmol/L compared to those between 50–75 nmol/L. Similarly, a randomised clinical trial of a range of vitamin D supplementation doses in women aged 57–90 years reported increased falls in participants with serum 25(OH)D levels above 112 nmol/L, whereas those with levels ranging between 79–95 nmol/L reported significantly fewer falls [11,15].

Thus, vitamin D appears to have a dual effect in which both too low and too high may be harmful, at least to the physical function and falls and fracture risk. However, the exact reason for this effect is unknown. Since muscle weakness and fatigue has been linked to low vitamin D, it has been popular to surmise that similar muscle function impairments are also responsible for the deleterious effects of high vitamin D. However, whether this is specifically a muscle effect, and whether it is the total amount of vitamin D or the rate at which it is increased has the greatest effect is unknown.

Given the number of inter-related confounding variables in human participants, as well as knowledge of the possible detrimental effects of high-dose vitamin D, we aimed to investigate the effects on the skeletal muscle of four weeks of very high daily oral vitamin D dietary supplementation in mice. This will be compared to a single bolus dose of vitamin D equivalent to the amount of vitamin D obtained in the diet over the four weeks.

## 2. Materials and Methods 

All experiments were approved by the Victoria University Animal Ethics Committee and conformed to the Australian code for the care and use of animals for scientific purposes (8th edition). Forty C57BL/6J female mice were purchased from the Animal Resources Centre (Perth, WA Australia) at seven weeks of age and were randomly allocated (four per cage) and housed in the Western Centre for Health Research and Education (WCHRE) Animal Holding Facility. After one week of acclimatisation, animals were placed on either a standard chow diet (AIN93G, Specialty Feeds, Glen Forrest, WA, Australia) containing 1000 IU/kg 25-hydroxyvitamin D3 (cholecalciferol) (CON; *n* = 8), or the otherwise same diet enriched with vitamin D (20,000 IU/kg cholecalciferol) (HIGH; *n* = 12). Another group (YEAR; *n* = 12) were injected with a bolus of cholecalciferol (1500 IU, Sigma), and then remained on the standard chow diet. Animals consumed the diets for four weeks, and were allowed ad libitum access to both chow and water, which were monitored three times per week. Based on an average animal food ingestion of 2.5 g/day, mice were expected to ingest the equivalent of ~1200 IU/year, which would equate to a human dose of ~1000 IU/day which is a typical supplementation dose. The HIGH diet was designed to provide this same yearly dose over the four weeks of supplementation, while that same yearly dose was provided in a bolus injection in the YEAR group. CON animals ingested less than expected (~2.3 g/day) over the four weeks, which equates to ingesting only ~850 IU per year if they maintained the same ingestion rate of that diet for the whole year (see Table 1). This equates to a human equivalent of ~770 IU/day (see Table 1), such that the CON animals on the standard diet ingested vitamin D comparable to typical human supplementation regimes. The animals in the HIGH group ingested their yearly dose in the four weeks of supplementation. In fact, since those animals actually ingested more of the high diet than the standard diet (see Table 1), this ended up being about 20% more than the initial target yearly dose of 1200 IU. As one of the primary aims was to compare high dose supplementation with a single bolus dose of vitamin D, the actual ingested dose of 1500 IU in the four weeks of supplementation in the HIGH group was replicated in the YEAR group. The YEAR group was injected by i.p. with a single dose of cholecalciferol dissolved in corn oil with a corn oil only vehicle group as control (*n* = 8).

At 12 weeks of age, animals were deeply anaesthetised with sodium pentobarbitone (60 mg/kg) and the extensor digitorum longus (EDL; fast-twitch) and soleus (slow-twitch) muscles were removed for contractile testing. Muscles were suspended in custom-built organ baths (ZULTEK Engineering, Melbourne, Australia) between a sensitive force transducer at one end and an immovable pin at the other flanked by field stimulating platinum electrodes. Muscle patency was maintained by a Krebs–Henseleit bicarbonate buffer bubbled with carbogen (5% CO_2_ in O_2_; BOC gases, Melbourne, Australia) at pH 7.4. Muscles were tested at 30 °C. Briefly, the muscle optimal length was established by a series of supramaximal twitch contractions (0.2 msec square wave pulse) delivered by an inbuilt stimulator coupled to an amplifier to ensure total motor unit recruitment. Following this, a force-frequency (F-F) relationship was established by stimulating the muscles with repetitive stimuli (train rates of 350 msec and 500 msec for EDL and soleus, respectively) at increasing frequencies. Peak isometric force (Po) was recorded as the peak tetanic force obtained during the F-F testing protocol. Muscle fatigue was produced by delivering repeated tetanic stimuli (100 Hz EDL and 80 Hz soleus) at a rate of one every four seconds for EDL and one every two seconds for soleus in an attempt to produce similar amounts of fatigue in both muscles. Recovery from fatigue was followed by delivering a single tetanic stimulation at various time points for 60 min. At the completion of testing, muscles were blotted dry, cut free from tendons and weighed. Peak force was normalised for muscle cross sectional area (CSA) according to the formula CSA = muscle mass / [optimal length x (fiber length: muscle length) x density] where the fiber length to muscle length ratio equates to 0.44 for the EDL and 0.71 for the soleus, and density 1.06 g/cm^3^ as previously described [17].

Data is displayed as mean ± SEM. Control data (CON and corn oil vehicle) was pooled with no differences between them and groups were analysed with a one-way ANOVA using SPSS v22 (IBM Australia Ltd, St Leonards NSW 2065, Australia) or Graphpad prism v7 (GraphPad Software, San Diego, CA 92108, USA).

## 3. Results

Body composition data can be seen in Table 2. The body mass of all groups was not different at 12 weeks after the intervention. Similarly, there were no differences in the absolute mass of the EDL or soleus muscles, although there was a trend for the soleus muscle to be lighter after the yearly injection compared to the other groups (*p* < 0.1). There were also no differences in the relative muscle mass between the groups.

The force-frequency relationship demonstrates the relative force generated from the muscle at a particular frequency of activation. In the EDL, YEAR animals produced significantly less force that both the CON and HIGH animals at 50 Hz and 80 Hz (*p* < 0.05, see Figure 1). Similarly, the YEAR soleus muscles produced significantly less force at 10–50 Hz compared to the CON animals (*p* < 0.05) and significantly less force at 30–50 Hz compared to the HIGH soleus (*p* < 0.05). No differences existed between the CON and HIGH animals, other than the HIGH animals producing more force at 30 Hz in the soleus muscle.

Absolute force generation by the EDL muscles was lower after the YEAR dose compared to the CON muscles (*p* < 0.05; see Table 1). However, the more gradual introduction of high doses of vitamin D through diet did not have this effect, such that EDL muscles from the HIGH animals was significantly greater than the YEAR animals (*p* < 0.05) and was not different from the CON. The same effects were observed in the soleus muscles, with YEAR muscles lower than both the CON and HIGH soleus muscles (*p* < 0.05), but with the addition that HIGH soleus muscles produced even larger forces than CON animals (*p* < 0.05, see Table 1).

With no differences in the calculated CSA of the muscles, similar effects were observed in the specific forces of the muscles (see Figure 2). In the EDL, the YEAR animals produced lower forces than either the CON or HIGH groups (*p* < 0.05), while in the soleus, although there was no difference in the soleus between the CON and YEAR animals, the HIGH soleus muscles produced significantly more force corrected for the cross sectional area than both other groups (*p* < 0.001).

There was no effect on fatigability, nor recovery from fatigue, in the EDL muscles between any of the groups (see Figure 3a). However, the YEAR soleus displayed an initial faster rate of fatigue (first minute) than either the CON or HIGH groups (*p* < 0.05). Similarly, the YEAR soleus also demonstrated a slower rate of recovery in the first two minutes post-fatigue (*p* < 0.05), and an overall lower recovery throughout (see Figure 3b).

## 4. Discussion

High bolus doses of vitamin D have the potential to increase the risk of falls, and it has been suggested that this could be due to effects on the skeletal muscle. The major finding of the current study was that injecting mice with a single bolus dose of vitamin D decreased the force produced at frequencies commonly used to activate muscles in vivo, compared with mice that consumed normal amounts of vitamin D. The bolus dose of vitamin D also decreased the relative force from the fast-twitch EDL muscle; muscles of this type would be expected to be activated during rapid activation in an attempt to avert falling after a trip [18,19]. Further, the soleus muscle demonstrated higher initial fatigability with repetitive stimulation, and slower recovery after the yearly injection; fatigue in a postural muscle could contribute to an increased risk of falls by limiting toe clearance when lifting the foot, for example. Interestingly, this is similar to associations between low vitamin D and increased fatigability [20]. Collectively, the data support the idea that muscular changes in response to a very large bolus of vitamin D could be harmful and contribute to the risk of falls observed in human participants. In contrast, the relative lack of effect of ingesting the same amount of vitamin D over a four-week period suggests that elevating vitamin D in a slower manner prevents the same detrimental effects.

This would be in agreement with a number of studies in which rapid elevations in vitamin D have resulted in deleterious effects on falls and fractures [11,13,14,15,21]. However, still using fairly high doses, but over a longer period, did not produce these same detrimental effects. For example, Trivedi et al. [12] in a double-blinded randomised controlled trial reported no adverse outcomes with 100,000 IU vitamin D given orally as a single capsule once every four months for five years in elderly participants. Further, the trial reported that the oral vitamin D supplementation may prevent or reduce fractures as the total fracture incidence was reduced by 22%, while a 33% reduction was observed in fractures in the most commonly broken osteoporotic sites.

Vitamin D deficiency has been associated with muscle weakness and fatigue [20]. As such, vitamin D supplementation is a common treatment. However, the current study is the first to provide supplementation when starting from a non-deficient baseline level (since mice had been consuming chow containing typical levels of vitamin D for eight weeks prior to the study commencing), thus avoiding changes that may have occurred in the muscle function as a result of vitamin D deficiency. Indeed, the study by Girgis et al. [22] demonstrated profound effects of depleted vitamin D levels on grip strength in mice. Importantly, the current study also investigated the isolated muscle function. It is likely that the effects of vitamin D deficiency are related to hormonal, neural and muscular effects. By investigating isolated muscles, whether the effects observed are due specifically to the skeletal muscle can be obtained. The YEAR dose deleteriously affected the muscle function by reducing the force at sub-maximal frequencies of activation and lowering the specific force. The former could suggest faster contraction time of the muscles, as vitamin D supplementation after vitamin D deficiency has been associated with higher proportions of type II fibers [23]. This would also be consistent with higher fatigability of the soleus muscle, as the soleus muscle of mice is a mixture of type I (slow-twitch) and type II (fast-twitch) muscle fibers [24], thus higher fast-twitch proportions would be expected to reduce fatigue resistance and slow recovery. The same fatigue effect may not be observed in the EDL muscle, as it is already virtually entirely composed of type II fibers. Fiber type effects have been observed in human studies, albeit in vitamin D deficiency. A randomized controlled study conducted by Sato et al. [9] on 96 elderly women with post stroke hemiplegia with serum 25-dihydroxycalciferol levels less than 25 nmol/L, investigated the effects of low-dose vitamin D on muscular atrophy. Forty-eight of the participants were given 1000 IU of ergocalciferol daily while the remaining received placebo over the course of two years. Patients that received the vitamin D dose were found to have an increase in the percentage of the type-II muscle fibers present and size. However, the mechanism of how this occurs remains uncertain, as to whether the relative number of type-II fibers had increased due to new formation or if existing type-I fibers were converted into type-II [9,25]. Given that type-II fibers are most predominantly used to prevent excessive mechanical stress or load during a fall or intense activity and in muscle strength training [25], it could be expected that more type-II fibers would in turn increase the muscle strength. However, that effect was not observed in the current study, as the YEAR group displayed significantly lower absolute and specific forces. Thus, providing a bolus dose of vitamin D caused both a lowering of force production and worsening of fatigue. Importantly, the same effect was not seen in the HIGH group, which demonstrated no effect on fatigue or recovery, and higher specific force productions. In a study in which muscle biopsies were obtained from 11 elderly patients with bone loss whom were treated with 1-alpha-hydroxycholecalciferol, an analog of vitamin D and calcium [23], histological classification revealed an increase in the percentage of fast-twitch type II A fibers and their cross-sectional area [23]. While more investigation would be required to demonstrate these same effects in the current study, it does appear that supplementation with very high doses of vitamin D eliciting gradual increases in serum vitamin D may improve muscle strength due to the increased numbers/proportion of type II fibers and mean diameter, which is not apparent with a single bolus of vitamin D.

Alternately, both alterations in the fiber activation and specific force in our study could be due to alterations in calcium handling of the muscle. Faster uptake of calcium could explain the rightward shift of the force-frequency curves, while reduced calcium sensitivity (or reduced calcium release from the SR) could contribute to reduced force per cross sectional area. However, no measures of calcium handling were performed, and thus the single muscle fiber analysis should be undertaken to help answer this question.

The exact reason for the increase in falls observed in high vitamin D dose studies [11,13,14,21] are not obvious as there are numerous factors in play. However, the current results have suggested that a rapid increase in vitamin D levels can cause muscle weakness specifically (as evidenced by decreased absolute and specific force). Toe clearance has been associated with tibialis anterior (TA) strength, and toe clearance is an important factor in avoiding falls [26,27,28,29]. Given that we have observed lower forces in two other hindlimb muscles, it is reasonable to assume that the same detriments would occur in the other hindlimb muscles, such as the TA. Similarly, increased falls are associated with muscle fatigue [30,31,32]. This study has also demonstrated faster fatigue and slower recovery in postural slow-twitch soleus muscles after a bolus yearly dose of vitamin D, thus making a contribution of fatigue to increased falls likely. Importantly, these same effects were not observed when the same dose was given to animals over a four-week period. Despite the very large doses, the more gradual introduction of the high vitamin D dose did not have the same detrimental effect. The reason for this is not immediately apparent but may be due to improved handling of vitamin D due to the altered vitamin D receptor number or sensitivity, or alteration in the main enzymes responsible for its intracellular action. Further work is required to elucidate these potential mechanisms.

There were several limitations to this study. Firstly, 25(OH)D was not actually measured in the plasma of the animals, so we do not know the exact levels of vitamin D achieved. However, the studies of dietary and high-injected doses have previously shown that elevations in vitamin D levels commensurate with the amounts given, both in humans and also rodents [33,34]. Indeed, mice fed the same dose (20000 IU/kg), albeit over a longer period, demonstrated elevated serum vitamin D. Likewise, i.p. injection (albeit in rats and over two doses) of a similar vitamin D dose as used in the current study has also demonstrated elevated vitamin D. As such, there is nothing to suggest that the plasma levels would not be reflective of total vitamin D intake/injection in the current study. In addition, histological analyses were not conducted thus alterations to the muscle fiber type and size could not be determined and should be considered in future studies. The vitamin D was given to young mice (eight weeks of age) which are still developing, and thus any effects on vitamin D receptors and muscle growth may not be reflective of what would happen in the aged muscles, which are generally the subject of vitamin D supplementation studies. As such, similar studies on older animals should be performed. Additionally, it is possible that injecting the vitamin D bolus i.p. may alter the vitamin D metabolism and/or have heightened effects in the muscle compared to if the same bolus was ingested orally, as the vitamin D diet was in the HIGH group. Using the oral gavage of the bolus dose would help to determine if the route of administration had any significant effect.

Notwithstanding the above, the current study suggests that the rapid elevation of vitamin D by a single i.p. bolus of vitamin D causes muscle dysfunction, particularly lower force output and slower recovery from fatigue of postural slow twitch muscles, compared with daily oral consumption of very high vitamin D diet. This may contribute to the increased risk of falls observed in studies in which bolus and intermittent high doses of vitamin D have been provided. Results of this study have implications on how vitamin D is delivered. Given that there may be good reason to try and increase vitamin D levels quickly, investigating ways in which this rapid elevation can occur without deleterious effects on muscles, such as in combination with increased physical activity should be pursued.

## Figures and Tables

**Figure 1 nutrients-11-01097-f001:**
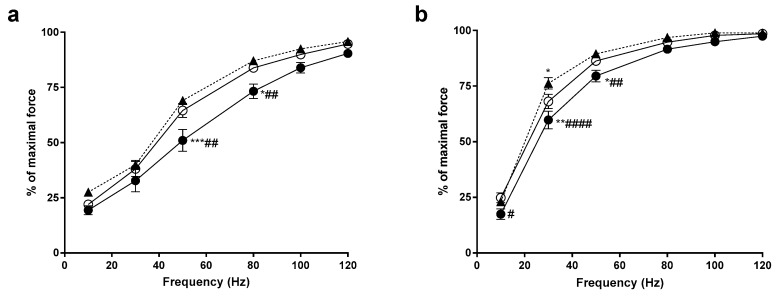
Force-frequency relationship of the extensor digitorum longus (EDL) and soleus (SOL) muscles. CON, open circles (○); HIGH, closed triangles (▲); YEAR, closed circles (●). (**a**) In the EDL, it was found that the force generated by the YEAR dose animals at 50 Hz and 80 Hz was significantly lower compared to both CON and HIGH animals; (**b**) in the SOL, the YEAR dose animals produced lower forces at 10 Hz, 30 Hz and 50 Hz compared to CON and HIGH animals. The force obtained by the HIGH animals at 30 Hz was significantly higher than the CON, however there were no differences observed above 30Hz. Symbols indicate: * *p* < 0.05, ** *p* < 0.01, *** *p* < 0.001; different from CON, # *p* < 0.05, ## *p* < 0.01, #### *p* < 0.0001; different from HIGH.

**Figure 2 nutrients-11-01097-f002:**
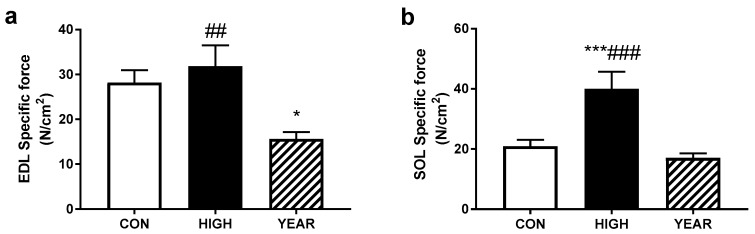
EDL and SOL specific force. (**a**) When the peak tetanic force was corrected for the force produced per cross sectional area (CSA), it was found that the YEAR animals displayed decreased force production in the EDL but is restored when administered in a high vitamin D diet over four weeks; (**b**) however, in the SOL, there was no effect found in the YEAR animals, instead the HIGH animals displayed increased force production when compared to both the CON and YEAR animals. Symbols indicate: * *p* < 0.05, *** *p* < 0.001; different from CON, ## *p* < 0.01, ### *p* < 0.001; different from YEAR.

**Figure 3 nutrients-11-01097-f003:**
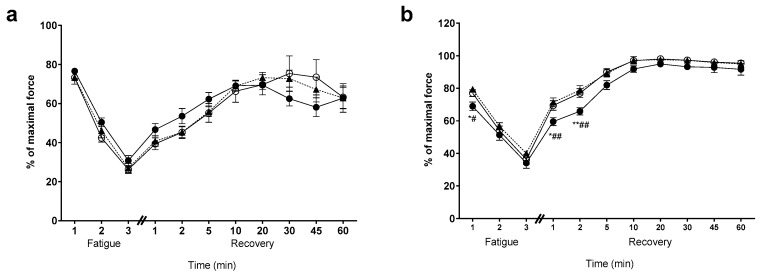
Fatigue and recovery for the EDL and SOL muscles. CON, open circles (○); HIGH, closed triangles (▲); YEAR, closed circles (●). (**a**) There was no effect of diet intervention or dose administration on the EDL fatigue and recovery from fatigue; (**b**) in the SOL however, it was found that the YEAR animals were slightly more fatigable during the first minute of fatigue when compared to the CON and HIGH groups; (**b**) the YEAR animals displayed delayed recovery at one min and two min when compared to the CON and HIGH groups. Symbols indicate: * *p* < 0.05, ** *p* < 0.01; different from CON, # *p* < 0.05, ## *p* < 0.01; different from HIGH.

**Table 1 nutrients-11-01097-t001:** Comparison of Vitamin D consumed.

	Diet (IU per kg Food)	Food Intake (g/d)	IU/Day	IU Consumed in 28 days	Yearly Equivalent ^+^ (IU)	Human Daily Equivalent * (IU/d)
CON (*n* = 8)	1000	2.31 ± 0.27	2.3 ± 0.2	65 ± 5	~840	~770
HIGH (*n* = 12)	20,000	2.74 ± 0.44	54 ± 7	1509 ± 202	~19,700	~18,000
YEAR (*n* = 12)			1500 ^#^	1500 ^#^		

+ assuming continued average daily intake. * based on a 20 g mouse compared to 80 kg human and metabolic scaling factor of 12 (see reference [16]). # single bolus dose.

**Table 2 nutrients-11-01097-t002:** Muscle morphometric data.

	CON (*n* = 16)	HIGH (*n* = 12)	YEAR (*n* = 12)
BODY MASS (mg)	20.6 ± 0.4	21.5 ± 0.6	20.5 ± 0.3
EDL MUSCLE MASS (mg)	9.23 ± 0.41	8.75 ± 0.25	8.52 ± 0.22
SOLEUS MUSCLE MASS (mg)	7.99 ± 0.39	7.87 ± 0.43	7.03 ± 0.38
EDL MUSCLE MASS: BODY MASS (mg/g)	0.45 ± 0.02	0.41 ± 0.02	0.42 ± 0.01
SOLEUS MUSCLE MASS: BODY MASS (mg/g)	0.39 ± 0.01	0.37 ± 0.03	0.34 ± 0.01
EDL P_o_ (mN)	448 ± 27	507 ± 59	267 ± 23 *^,#^
SOLEUS P_o_ (mN)	204 ± 17	331 ± 37 *	156 ± 16 *^,#^
EDL CSA (mm^2^)	1.73 ± 0.11	1.55 ± 0.07	1.68 ± 0.06
SOLEUS CSA (mm^2^)	0.98 ± 0.04	0.92 ± 0.07	0.89 ± 0.04

Po = peak tetanic force; CSA = cross sectional area. * significantly different from CON, #significantly different from HIGH.

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
