# Peer review of "The Effect of Yearly-Dose Vitamin D Supplementation on Muscle Function in Mice"

_nutrients, 2019, doi:10.3390/nu11051097_

Reviewer 1 Report

High dose, bolus vitamin D has now been shown by several researchers, as you have discussed, to increase falls in elderly humans.  Investigating a murine model and comparing standard vitamin D dosing with high dose daily and a single bolus Vitamin D doses, equivalent to those reported to be used clinically was a robust design.  You chose an "ex vivo" muscle contraction model, which allowed you to test muscle responses in isolation from "in vivo" whole tissue including neurovasculature bundles and adjacent bone, which as you highlight has both strengths and weaknesses with regard to research design.  You acknowledge lack of histology and plasma 25 OH vitamin D measurements as minor limitations of your study, which is probably reasonable.  In lines 248 - 253 of the discussion, you allude to calcium fluxes as having a role in muscle function, but did not venture the underlying mechanism for this as relates to Vitamin D.  It may instead be better to consider replacing this paragraph with a discussion regarding extra-renal vitamin D metabolism, which is unregulated by PTH, IGF 1 and other hormones as with renal 1 alpha hydroxylase, and consequently 1,25(OH)2 vitamin D generation is simply dependent on 25 OH vitamin D levels.  Is there evidence of 1 alpha hydroxylase within this murine mouse model and are there candidate genes with vitamin D response elements, which may be considered as having a role in the muscle changes with high dose vitamin D? 

Some minor points, line 110, please insert lower case"2" for CO2 and O2;  line 228 the 25 OH vitamin D concentration of 10ng/ml please also convert to SI units.

Reviewer 2 Report

This manuscript investigates bolus of vitamin D supplementation on muscle function in mice by estimating the force-frequency relationship and peak isometric force of extensor digitorum longus and soleus muscles. The authors concluded that large bolus doses of vitamin D significantly less force-frequency relationship in both muscles, however, long-term supplementation of 20 times the normal dietary dose of vitamin D increased peak tetanic force in extensor digitorum longus and soleus muscles. For the results, the author puts forward some hypotheses but does not do mechanism research.

Do the authors have data about serum the vitamin D level?

Mini comment:

I did not find in reference 17 to mention the fiber length and density as the authors mentioned.
